

# Dynamic hysteresis across a dissipative multi-mode phase transition

Ⓘ **Marvin Röhrle**, Ⓘ **Jens Benary**, Ⓘ **Erik Bernhart** and Ⓘ **Herwig Ott**⋆

Department of Physics and Research Center OPTIMAS,
University of Kaiserslautern-Landau, 67663 Kaiserslautern, Germany

⋆ ott@physik.uni-kl.de

## Abstract

Dissipative phase transitions are characteristic features in open quantum systems. Key signatures are the dynamical switching between different states in the vicinity of the phase transition and the appearance of hysteresis. Here, we experimentally study dynamic sweeps across a first order dissipative phase transition in a multi-mode driven-dissipative system. In contrast to previous studies, we perform sweeps of the dissipation strength instead of the driving strength. We extract exponents for the scaling of the hysteresis area in dependence of the sweep time and study the $g^{(2)}(0)$ correlations, which show non-trivial behavior. By changing the temperature of the system we investigate the importance of coherently pumping the system. We compare our results to numerical calculations done for a single mode variant of the system, and find surprisingly good agreement. Furthermore, we identify and discuss the differences between a scan of the dissipation strength and a scan of the driving strength.

# 1   Introduction

Open quantum systems are often a more faithful representation of real world systems as compared to a unitary Hamiltonian description of a closed and isolated system. At first glance, the interaction with the environment, which is a native property of open quantum systems, seems to be a nuisance as it introduces dissipation (exchange of particles and energy) and decoherence. However, at second glance, the possible engineering of the environment allows for the robust preparation of quantum states in the presence of decoherence and dissipation [1,2]. Thereby, the intrinsic attractor dynamics provides the means to create such quantum states irrespective of the initial state. Moreover, open systems exhibit new phenomena without a corresponding counterpart in the unitary quantum regime, for instance dynamical switching between different system configurations. A particularly important class of states in open systems are steady-states, whose density operator does not change in time.

In open quantum systems the local conservation of energy and particles is in general no longer valid, since there is a constant exchange with the environment. As a consequence, well established equilibrium concepts have to be adapted for open quantum systems. Most notably, the notion of phase transitions has to be redefined for open systems [3,4]. Many open quantum systems under investigation are driven-dissipative systems. There, an additional gain mechanism counteracts the dissipation and the steady-states show interesting emergent properties. In fact, most transport phenomena in physics are related to the coupling to corresponding reservoirs and are thus best described as an open system. If the properties of such a steady-state change in a non-analytical way upon a change in the system parameters, this is referred to as a dissipative phase transition [4].

Driven dissipative systems are often encountered in photonics. They exhibit metastable states, which can be used for rapid switching, e.g., in lasers [5] and in optical cavities [6,7]. From early on, optical bistability [8] was found to be a foundational example for a first order dissipative phase transition. Further examples of dissipative phase transitions with quantum gases include cavity QED [9–11], Bose-Einstein condensates in optical lattices with local dissipation [12], and photon Bose-Einstein condensates [13].

For most studied systems it is only possible to dynamically change the driving strength, since the dissipation strength is given by external constraints. However, there are methods to engineer losses in a controlled way, e.g., via photoassociation [14], (near) resonant light [15–17], or Raman coupling to auxiliary states [18]. Hysteresis with driving strength sweeps has been studied in many systems, some closely related one are in cold atom systems [19–22], Rydberg atoms [23–25], or exciton polariton condensates [26,27]. There have been no studies of hysteresis behavior by controlled change of dissipation. In this work we want to address this open question by making dissipation strength sweeps over a first order dissipative phase transition for a fixed drive. We are in particular interested in the similarities and differences to the sweeps of the driving strength.

# 2   Physical setup and theoretical description

We realize the driven-dissipative system with an ultracold quantum gas of bosonic $^{87}$Rb atoms in a one-dimensional optical lattice. Each lattice site contains a pancake-shaped quasi 2D BEC of about 800 atoms. All lattice sites are coherently coupled to each other with tunneling coupling $J$. The radial trap frequency is small compared to the chemical potential of each condensate, which makes each pancake a multi-mode system with more than 80 transverse harmonic oscillator modes occupied. A focused electron beam is used to remove atoms from a single lattice site. Thereby, the electron beam ionizes the atoms, which are then detected

by a channel electron multiplier as a time dependent ion signal $I(t)$, which is proportional to the occupation of the site. This weak probing scheme allows us to continously monitor the system while creating a local dissipative process. The loss/dissipation rate $\gamma$ is set by the intensity of the electron beam. Since the beam diameter of the electron beam is 150 nm full width half maximum, it can resolve a single lattice site. To create a homogeneous loss over the whole transverse extent, we scan it periodically over the site with high frequency. Tunneling of atoms from neighboring sites into the lossy site consitutes the drive. The steady-state of the system is then determined by the competition between the losses and the inward flow of atoms. This system exhibits a first order dissipative phase transition, which we have analyzed in a precursor study [12] In Fig. 1 we present a sketch of the experiment. More detail about the experimental sequence for the dissipative hysteresis can be found in the supplementary material.

The studied system can be described theoretically with a multi-mode extension of the seminal Kerr model [8] (all equations with $\hbar = 1$)

$$\hat{H}_{\text{MM}} = \sum_i n_i \omega \hat{a}_i^\dagger \hat{a}_i + \sum_{ijkl} \frac{U_{ijkl}}{2} \hat{a}_i^\dagger \hat{a}_j^\dagger \hat{a}_k \hat{a}_l + \sum_i \left( J_i^* \hat{a}_i e^{i\mu t} + J_i \hat{a}_i^\dagger e^{-i\mu t} \right). \tag{1}$$

The first term describes the occupation of the transverse harmonic oscillator modes in the central site (energy $n_i \omega$), which we chose as a basis for the expansion of the atomic field. The creation and anihilation operators of each mode are denoted by $\hat{a}_i^\dagger$ and $\hat{a}_i$. The second term is the interaction of the particles in different modes with the interaction matrix elements $U_{ijkl}$. It allows for a redistribution of the atoms across different modes. The last term is a coherent drive with amplitudes $J_i$ due to tunneling of atoms to and from neighboring sites. The driving frequency is given by the chemical potential $\mu$ of the neighboring sites. Note that the tunneling coupling $J_i$ is different for each transverse mode, as the Franck Condon overlap in the tunneling matrix element also depends on the transverse shape of the wave function [12].

The above Hamiltonian is impossible to solve exactly due to the large number of atoms and modes and the all-to-all connectivity of the interaction matrix elements. Recently, there have been two theoretical studies to treat this multi-mode system. One employs a computationally expensive c-field method [28], which can reproduce many of the core results even quantitatively. The other combines the truncated Wigner approximation with a variational ansatz [29], thus being able to reproduce the system dynamics in the absence of dissipation. However, recent progress with a 1D Bose-Hubbard chain with a limited amount of sites is able to qualitatively reproduce the dynamical behavior as reported in earlier studies [30]. Therefore, we consider the well-known single mode version of the system's Hamiltonian, since it can be used for a qualitative comparison:

$$\hat{H}_{\text{SM}} = \omega \hat{a}^\dagger \hat{a} + \frac{U}{2} \hat{a}^\dagger \hat{a}^\dagger \hat{a} \hat{a} + F(t)^* \hat{a} e^{i\mu t} + F(t) \hat{a}^\dagger e^{-i\mu t}, \tag{2}$$

where we allow explicitly for a time-dependent external drive $F(t)$. To discern between single mode theory and multi-mode experimental results, the driving strength in the single mode models is written as $F$, whereas in the multi-mode case as $J$.

The time evolution is done via a standard master equation in terms of density matrices $\hat{\rho}$ and a Liouvillian super operator

$$\mathcal{L}\hat{\rho} = \frac{\partial \hat{\rho}}{\partial t} = i\left[\hat{\rho}, \hat{H}_{\text{SM}}\right] + \frac{\gamma(t)}{2}\left(2\hat{a}\hat{\rho}\hat{a}^\dagger - \hat{a}^\dagger \hat{a}\hat{\rho} - \hat{\rho}\hat{a}^\dagger \hat{a}\right). \tag{3}$$

The particle losses are described by the dissipation rate $\gamma(t)$, which we also allow to explicitly depend on time. This single mode master equation can be treated numerically for small occupation numbers $\hat{n} = \hat{a}^\dagger \hat{a}$.

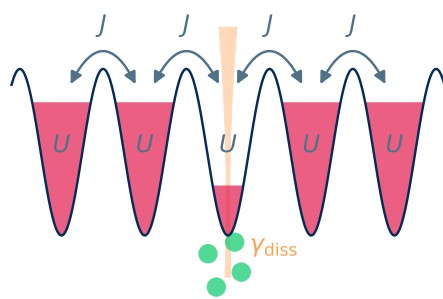

Figure 1: Sketch of the experimentally studied system. A 1D chain of quasi 2D condensates of rubidium atoms is coupled by the tunneling $J$. One site is subject to a local loss process, realized by a scanning electron microscope beam, which removes atoms from the cloud and ionizes them. Some of the ions are subsequently detected and are used as a time resolved measurement signal. Atoms from the neighboring sites can tunnel into the lossy site. More details on the experimental setup can be found in Ref. [12].

For the sake of completeness, we also calculate relevant quantities in mean-field approximation. This neglects quantum fluctuations and replaces all operators with complex valued fields. This results in the mean-field equation

$$\mathrm{i}\frac{\partial \alpha}{\partial t} = \left( \omega_c - \mathrm{i}\frac{\gamma(t)}{2} + U|\alpha|^2 \right)\alpha + F(t)\mathrm{e}^{-\mathrm{i}\omega_p t}, \tag{4}$$

whose time evolution can be computed by numerically integrating the complex valued function $\alpha$ for some initial conditions. The mean particle number in the system is given by $n(t)=|\alpha(t)|^2$. To study hysteresis in the system, either the dissipation strength $\gamma$ or the driving strength $F$ can be varied over time. Note that functional appearance of the two in Eq. 4 is different. Since the mean-field treatment does no longer explicitly treat quantum effects, it becomes less accurate for small particle numbers. For large occupation numbers instead, it can be a good approximation that agrees with the full quantum treatment of the system [31].

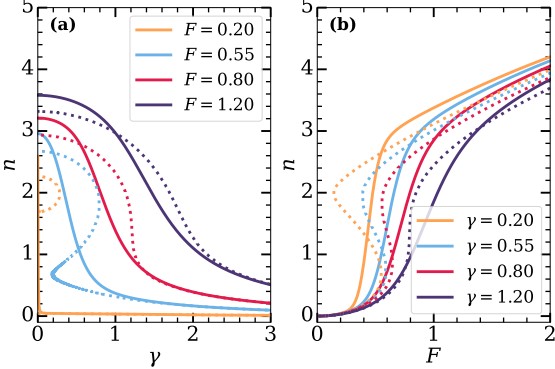

Figure 2: Steady states of the mean-field model (dotted lines) and master equation (solid lines) when varying $\gamma$ or $F$ for four different driving strengths $F$ or dissipation strengths $\gamma$ in (a) and (b), respectively.

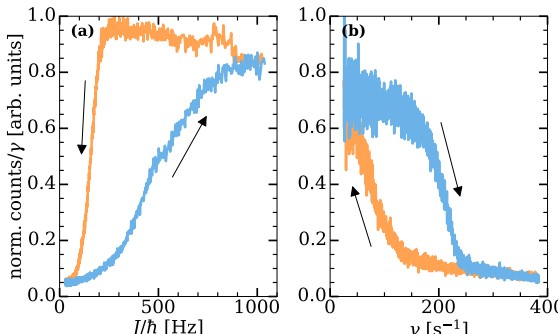

Figure 3: (a) Hysteresis measurement for a constant dissipation strength $\gamma = 340\,\text{s}^{-1}$ and a linear ramp in the lattice depth, starting with a deep lattice. One direction of the sweep takes $\tau_s = 160\,\text{ms}$ and the hysteresis is averaged over 415 runs. Both parts of the hysteresis are normalized individually. (b) Hysteresis for a constant tunneling coupling $J/\hbar = 290\,\text{Hz}$ and a linear ramp in the dissipation strength $\gamma$, starting with a weak dissipation. The sweep time is $\tau_s = 500\,\text{ms}$ and we average over 142 runs.

## 3 Results

Before we consider the case of dynamically changing a system parameter, let us first theoretically look at the steady states for fixed parameters in order to highlight key differences, when varying the dissipation instead of the driving strength. In Fig. 2 we compare the steady-state for both cases calculated with the master equation and the mean-field approximation. Since the mean-field equation Eq. 4 can not be analytically solved with respect to $\alpha$ when varying $\gamma$, the implicit equation is solved numerically to obtain the steady states. For the master equation, we perform sparse matrix decomposition to obtain the steady state matrix [32]. We then calculate the expectation value for the number operator $\hat{n} = \hat{a}^\dagger \hat{a}$ with the steady state density matrices. Bistability in the mean-field occurs for certain parameter ranges and the characteristic "S"-shape is recovered, however, in the case of varying the dissipation strength $\gamma$ it resembles a "Z"-shape. For the master equation, the expectation value is unique in all cases and no signs of bistability occur in either case [8]. However, the steady state $\hat{\rho}_{ss}$ can be a mixture of the mean-field steady states and the system switches between the two [33]. A noticable difference is that for vanishing dissipation, the mean-field still predicts two stable solutions for small enough drive (Fig. 2(a)). Instead, when keeping the dissipation fixed and changing the drive (Fig. 2(b)), the system always leads to the trivial solution $n = 0$ case for $\gamma > 0$ for small $F$. Only when $\gamma = 0$, bistability prevails down to $F = 0$. While the basic characteristics are rather similar, one might expect also some differences in the dynamic hysteresis behavior, especially in the limit of small drive and dissipation strength.

### 3.1 Hysteresis measurement

For measuring the hysteresis, we either prepare the system initially with the same occupation number of the central site as compared to the neighboring sites or we remove the atoms from the central site with the electron beam prior to the experiment, thus starting form an empty site. We then sweep either the dissipation strength $\gamma(t)$ or the tunneling coupling $J(t)$ across the phase transition back and forth for a variable sweep time $2\tau_s$. The time dependent ion signal, which is proportional to the number of atoms occupying the central site, is the basis for the dynamic observation of the system. For each parameter set, we repeat the experiment many times to measure the expectation value and the fluctuations of the atom number.

In Fig. 3, we show the two different types of hysteresis. In both cases, a large hysteresis area is apparent. This confirms that the qualitative behavior for both types of sweeps is the same. Varying the dissipation strength is therefore an equally well suited approach to characterize the hysteresis at a dissipative phase transition as varying the drive strength. In the following, we characterize the hysteresis area and its scaling with the sweep time as well as the fluctuations of the atom number during a hysteresis scan.

## 3.2 Atom number fluctuations

Hysteresis measurements typically concentrate on the evolution of the expectation value of the order parameter of the system. Here, we can also look at the atom number fluctuations across the hysteresis. In Fig. 4 we show the normalized two-body correlation function

$$g^{(2)}(0) = \frac{\langle I^2 \rangle}{\langle I \rangle^2} - \frac{1}{\langle I \rangle}, \tag{5}$$

where $I$ are the ion counts in the central site for a given time bin and $\langle \cdot \rangle$ is the ensemble average over many realizations. In case of no fluctuations beyond Poissonian noise, one would expect a constant value of 1 for the whole protocol, which would be the case for a pure condensate. However, we find that within the steep slopes of the hysteresis area, the $g^{(2)}(0)$ value becomes much larger than 1. This indicates large shot to shot fluctuations in the individual realizations of the experiments, corresponding to the system jumping at slightly different dissipation strengths between the two mean-field bistable branches. The atom number fluctuations are therefore pronounced and a direct consequence of the underlying switching dynamics in the vicinity of the phase transition. Furthermore, for dissipation strengths smaller than the positions of the two peaks, when the system is in a superfluid state, the $g^{(2)}(0)$ value is around 1 (dashed line in Fig. 4(a)). In the case of larger dissipation strengths, $g^{(2)}(0)$ is slightly larger than 1 (dotted line in Fig. 4(a)). This reflects the superfluid nature of the steady-state for small dissipation strengths (high filling), which suppresses density flucutations, while the atoms for high dissipation strengths are in a normal state (low filling), where bosonic bunching of different modes is present [34]. However, as many modes are involved, the value does not reach $g^{(2)}(0) = 2$ [35]. The corresponding calculation for the single mode system is shown in Fig. 4(b). The fluctuations in the single mode model show similar behavior and the two maxima have roughly the same magnitude. For small $\gamma$ the blue curve in Fig. 4(b) even exhibits slight anti-bunching, which can be attributed to interference of a coherent reference with the output from a non-linear medium [36]. However, in the experimental data there is no sign of anti-bunching. A possible explanation might be the multi-mode nature of the system, which allows for the formation of a condensate with Poissonian statistics.

## 3.3 Scaling of the hysteresis area

Dynamic hysteresis measurements depend on the sweep time, since for an infinitely slow sweep, the system would always be in its steady state and hysteresis would be absent. An important quantity is the hysteresis area $A$,[1] which should vanish for $\tau_s \to \infty$. The experimentally measured scaling of the area is shown in Fig. 5(a), where an increase for shorter sweep times is clearly visible. The exponent is extracted with a fit to a power law function

$$A \propto \tau_s^\alpha, \tag{6}$$

which directly yields the exponent $\alpha$. We find an experimental value of $\alpha = -0.87(4)$, when sweeping the dissipation strength, where the error is the covariance estimate of the fit.

---

[1]See Appendix C for the definition of the hysteresis area.

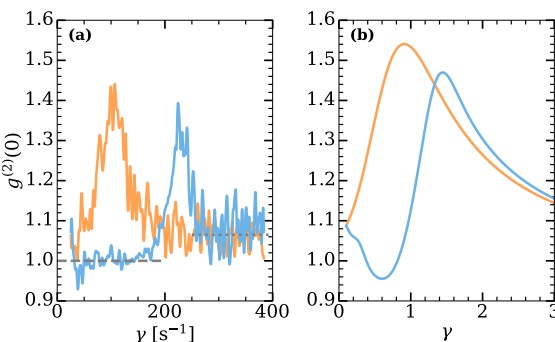

Figure 4: Normalized pair correlation function $g^{(2)}(0)$ for a hysteresis when seeping the dissipation. (a) Experimental data corresponding to the hysteresis shown in Fig. 3(b). A Gaussian filter is applied to make the behavior better visible. The dashed and dotted lines are a guide to the eye for small and large dissipation strengths. For further discussion see text. (b) Single mode master equation calculation with driving strength $F = 0.7$ and sweep time $\tau_s = 80$.

Theoretically, only the case of sweeping the driving strength $F(t)$ has been studied so far. In mean-field approximation an exponent of $A \propto \tau_s^{-2/3}$ has been found, when sweeping the driving strength $F$ [37]. When doing a similar calculation with a sweep of the dissipation strength $\gamma(t)$ instead, one can find a mean-field exponent of $A \propto \tau_s^{-1}$.[2]

If one includes quantum fluctuations via the master equation, a sweep of the driving strength in the limit of long sweep times yields a scaling of $A \propto \tau_s^{-1}$ [38]. The same scaling law has been found in other Kerr-type systems, e.g., by a geometric approach [39] or in a Kerr resonantor coupled to an ancilla two-level system [40].

To theoretically investigate the influence of quantum fluctuations for our case of a dissipation sweep, we have solved the master equation for the single mode system (Eq. 3). The results are shown in Fig. 5(b). For all studied driving strength values $F$, we find a scaling exponent of $\alpha = -1$. Thus, in contrast to a sweep of the driving strength, the mean-field and the full quantum description make the same prediction for a sweep of the dissipation strength.

Our experimentally measured exponent of $\alpha = -0.87(4)$ is slightly smaller than the theoretical prediction for the single mode system. However, the red dashed line in Fig. 5(a) with an exponent of $\alpha = -1$ is also an acceptable fit to the data. In order to get a better result, the hysteresis has to be measured for longer sweep times, which is not possible in our system, due to the constant atom loss.

## 3.4 Temperature dependence of the hysteresis

We now look into the importance of coherently pumping the system by changing the temperature of the whole sample. By increasing the temperature, the condensate fraction $n_c = N_c/N$ diminshes [41]. To investigate this regime, we initially prepare a cloud with a higher temperature and corresponding smaller condensate fraction $n_c$. In Fig. 6, we show the hysteresis area for different condensate fractions when changing the dissipation strength $\gamma(t)$ for a fixed $J$. Indeed, decreasing the condensate fraction leads to a decrease in the hysteresis area. A closer look at the plot reveals a kink at around $n_c = 0.4$, where the slope of the data changes. The origin of this effect is unclear.

We have also analyzed the behavior of the $g^{(2)}(0)$ correlations for smaller condensate fraction. We find that the two peaks in the $g^{(2)}(0)$ correlation function vanish for decreasing

---

[2]See Appendix B for the mean-field calculations of the area scaling law.

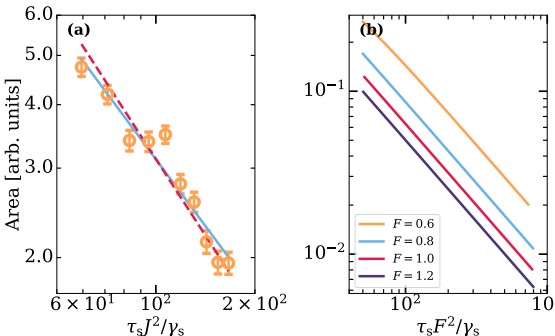

Figure 5: Scaling of the hysteresis area when sweeping the dissipation strength for different sweep times $\tau_s$. (a) Experimental results for different sweep times. The blue line is a power law fit with an exponent of $\alpha = -0.87(4)$. The dashed red line is a power law fit with a fixed exponent of $\alpha = -1$. The hysteresis protocol is the same as in Fig. 3(b). (b) Theoretical results obtained by density matrix time evolution of the master equation. For slow sweeps we find an exponent of $\alpha = -1$, irrespective of the driving strength $F$.

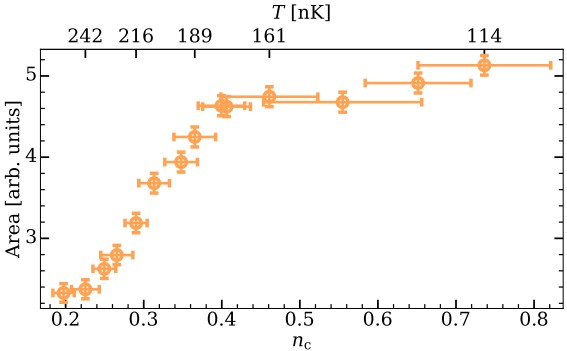

Figure 6: Hysteresis area at finite temperature. We show the hysteresis area in dependence of the temperature before loading the atoms into the lattice. Note that the number of atoms increases for an initially hotter sample. The hysteresis protocol is the same as in Fig. 3(b). The condensate fraction $n_c$ is given for certain data points individually.

condensate fraction. This can be interpreted as the transition between two distinct long-lived steady states for a coherent drive, where the system can only be in one of them, to a thermal state with vanishing hysteresis and normal fluctuations. In fact, when replacing the coherent drive in Eq. 2 with a coupling to a thermal bath in Eq. 3 the hysteresis also vanishes in the single mode model. We conclude that only with coherent coupling to the neighboring sites the system exhibits hysteresis when varying the dissipation strength $\gamma$, while, when reducing the coherent fraction, the hysteresis area starts to disappear.

## 3.5 Open hysteresis loops

In the discussion of Fig. 2, we have already noticed that in the limit of $\gamma \to 0$, the mean-field solution does not always become monostable. We now analyze the consequence of this for the hysteresis measurement. To this end, we study the behavior of the hysteresis area for a sweep of the dissipation strength with constant sweep time but varying tunneling coupling $J$ for an initially full site. The results are shown in Fig. 7(a). First, we find that increasing the tunneling coupling between sites increases the hysteresis area. Second, we find that the

SciPost Phys. **16**, 158 (2024)

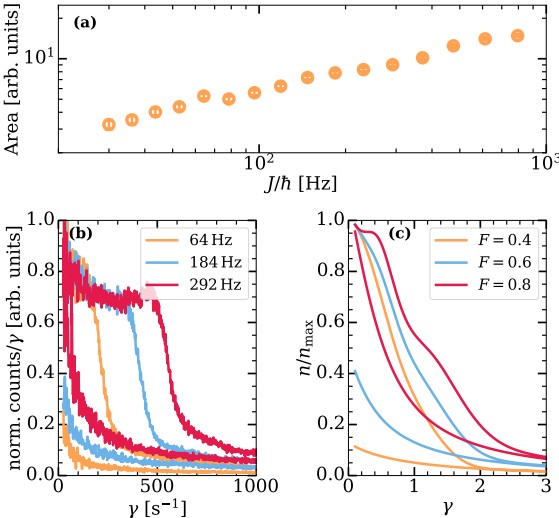

Figure 7: (a) Hysteresis area in dependence on the tunneling coupling $J$ for sweeps of the dissipation strength. In the hysteresis protocol, we linearly change the dissipation strength $\gamma$ from 70 Hz to 1800 Hz and $\tau_s = 400$ ms and back, starting from an initially full site. (b) Example of three hysteresis curves for different $J/\hbar$. They correspond to three data points of (a). The blue and the orange hysteresis loops do not close. (c) Numerical calculations of the single mode system in a similar parameter regime as the experiment. All hysteresis curves are normalized to 1.

hysteresis does not close anymore, if the tunneling coupling is too weak (orange and blue curve in Fig. 7(b)). At first this seems counter-intuitive, however, to understand the reason behind these two phenomena, let us look at each step of the hysteresis and the reaction of the system. When the system is in a state of high occupation number and low dissipation strength, it exhibits coherent perfect absorption [42]. All incoming matter waves are being absorbed and the occupation number stays constant. Increasing the dissipation strength will reach a regime where this effect breaks down. For large driving strengths this break down happens at larger values of the dissipation. Once the system is in the lower branch and the dissipation strength is ramped back again, it does not fully switch back to the high occupation state for small dissipation strengths $J$ (orange and blue curve in Fig. 7(b)) and the hysteresis loop does not completely close. As the refilling behavior is roughly the same for all $J$, this results in an increase of the hysteresis area with $J$.

The non-closing of the hysteresis curve is also reproduced by the theoretical model for the single mode system (Fig. 7(c)). However, it shows a decrease in hysteresis area for larger $J$. This can be viewed as manifestation of macroscopic quantum self-trapping, where the occupation number of the emptied site stays low even for vanishing dissipation strength, due to the difference in chemical potential [43]. In the multi-mode system at hand, macroscopic self trapping is not possible since the transverse mode splitting is smaller than the chemical potential and resonant tunneling is always possible at the cost of a reduced Franck Condon overlap, which leads to $J_i < J$ [44].

## 4 Summary and Conclusions

To summarize this paper, we have experimentally investigated dynamic hysteresis effects at a first-order dissipative phase transition of a multi-mode quantum system by sweeping the

dissipation strength $\gamma$. We find an asymptotic area scaling exponent slightly smaller than the one predicted by numerical calculations for a full quantum single mode version of the system. At the edges of the hysteresis area, we find pronounced density fluctuations from one realization to another. The hysteresis behavior is less pronounced for reduced condensate fraction and - depending on the parameter settings - the hysteresis does not necessarily close.

The observed phenomenology can be explained on a qualitative level with a single mode model. On a quantitative level, however, the multi-mode nature of the system becomes apparent. For example, the hysteresis areas and the $g^{(2)}(0)$-correlations show significant deviations. Also, only the multi-mode system features a condensation mechanism and bypasses the self-trapping mechanism, which is present in the single mode system. A quantitative modeling of the experiment would require an advanced theory as developed, e.g., in Ref. [28].

Our studies extend previous work on hysteresis scaling to many modes and stronger interactions. As the experiments are performed in a regime where exact numerical calculations do not work anymore, they can help to benchmark many-body theories in the vicinity of dissipative phase transitions. For the future, it would be important to develop a better understanding of the universal scaling laws at a dissipative phase transitions - very much in the same way as it is done with great success for equilibrium phase transitions.

## Acknowledgments

We would like to thank C. Kollath, M. Fleischhauer, and M. Davis for helpful discussions.

**Funding information**  The authors acknowledge financial support by the DFG within SFB/TR 185 OSCAR, project number 277625399.

## A   Experimental sequence

The main object of interest in this work are dissipative hystereses, which dynamically change the dissipation strength over time. An electron beam is used to realize the dissipative process, since it removes atoms from the cloud by ionizing them and we measure the time dependent ion signal $I(t)$, which is proportional to the number of atoms in the site. Therefore, the central site is subject to an effective dissipation strength $\gamma$. On the other hand there is the drive created by atoms tunneling from the neighboring sites of the 1D lattice with a tunneling coupling $J$. The lattice laser has a wavelength of 774 nm, i.e., it is blue detuned with respect to the $^{87}$Rb D2 line at 780 nm. We adjust the power of the lattice laser, which results in different lattice depths $s$ measured in the recoil energy $E_r = \hbar^2 k_{\mathrm{lat}}^2/2m$ and this also changes the tunneling coupling $J$. On the other hand varying the dissipation strength is done by changing the width of the scan pattern. The electron beam is constantly scanned over one site in 300 µs per line and by increasing the width beyond the position of the atoms, the effective dissipation strengths is reduced, since for some time the beam does not interact with the atoms. To measure the dissipative hysteresis, the width of the scan is varied from one line to another in order to dynamically adapt the dissipation strength. The timing of the sequence is shown in Fig. 8, which shows both sequences for an initially full or empty site. The first step is exponentially ramping up the lattice to $s = 20\,E_r$, which is high enough, that we can empty the site in 3 ms, when we prepare an initially empty site by strong dissipation for a short duration. For an initially full site, the electron beam is not switched on during this period, however, the lattice is still ramped up to have the same lattice depth changes in both cases. During the next 2 ms the lattice power is reduced while keeping the dissipation at the same strength. This ensures,

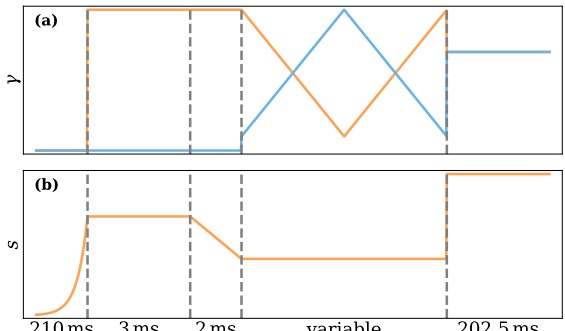

Figure 8: Sketch of the main experimental sequence after preparing a BEC for a dissipative hysteresis. Note that the x-axis, which represents the duration of each step is not to scale. The variable experimental duration corresponds to $2\tau_s$, which is the whole hysteresis duration. (a) The orange line corresponds to a measurement staring with an empty site and high dissipation and the blue line to a full site and low dissipation. (b) Lattice depth $s$ over the experimental sequence. For further information about each segment refer to the text.

that the initial occupation for the experiment is maintained during the ramp. Then the actual hysteresis part of the experiment starts. The dissipation strength is linearly changed across the whole range and after $\tau_s$ the dissipation strength is again linearly changed to the initial value over a duration of another $\tau_s$. After the hysteresis an image of a larger part of the lattice is taken to verify the position and final relative filling. This is done by scanning the electron beam in a rectangular area line by line over the atoms. The image is taken for a lattice depth $s = 30\,E_r$ which is enough to freeze out any remaining movement and keep the atoms in their respective sites for the whole imaging duration.

In case of the driving strength hysteresis, the initial two steps are the same, however, in the 2 ms the dissipation strength is ramped down or up for an initially empty or full site respectively. The lattice depth $s$ ramps to the initial value of the hystersis measurements. For the hysteresis the lattice depth $s$ is changed linearly in time with the same protocol as described for the dissipative hysteresis.

## B   Mean-field results

In mean-field approximation, we calculate the scaling laws of the hysteresis area for large sweep times in the following way. We integrate Eq. 5 for some initial condition of $\alpha$. Unless otherwise stated, we use $\gamma = 1$, $F = \gamma$, $\Delta = -2\gamma$, and $U = \gamma/2$. We start with $\gamma_1 = 5$ and linearly ramp it down to $\gamma_2 = 0.1$ and back up again. This results in a total range of $\gamma_s = |\gamma_1 - \gamma_2| = 4.9$. As initial condition for the population, we use $\alpha = 0$, since for a high dissipation strength the system should have an occupation close to zero. The resulting areas for different driving strengths $F$ are shown in Fig. 9.

For the five different driving strengths shown in Fig. 9, we find a scaling exponent for large sweep times of $-1$. This deviates from the result for a sweep of the driving strength, which was found to scale with an exponent of $-2/3$ [37]. For further comparison to the master equation solution of the single mode model, refer to the main text.

## C Hysteresis area calculation

The hysteresis area througout this whole work is defined as

$$A = \left| \int_{t < \tau_s} I(t) \mathrm{d}t - \int_{t > \tau_s} I(t) \mathrm{d}t \right|. \tag{C.1}$$

Here, $I(t)$ with $t \in [0, 2\tau_s]$ is the time dependent ion signal which is proportional to the occupation number in the central site. For the numerical simulations we use $n(t) = |\alpha(t)|^2$ for the mean-field solution and $n(t) = \langle \hat{n}(t) \rangle$ for the master equation calculations.

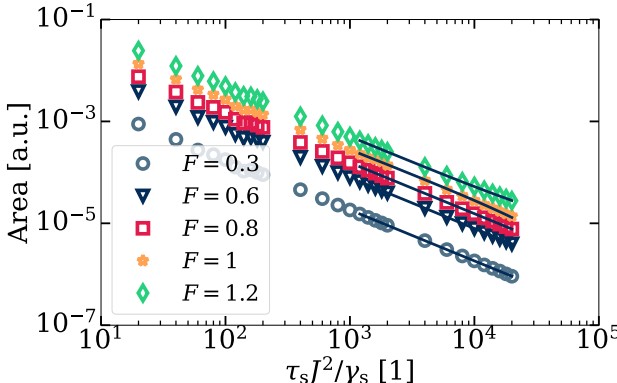

Figure 9: Hysteresis area scaling when sweeping the dissipation strength for an initially high dissipation strength for different sweep times $\tau_s$, see text. Five different driving strengths $F$ are shown. The lines correspond to power law fits, each yielding an exponent of $-1$.

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
