# Peer review of "Dynamic Hysteresis Across a Dissipative Multi-Mode Phase Transition"

_SciPost Physics, doi:SciPost Phys. 16, 158 (2024)_

## Round 1 · Referee Report · Anonymous (Referee 1) · 2024-2-15

Report

This paper presents theoretical and experimental investigations of the dynamics of a driven dissipative system dynamically crossing a first order dissipative transition. The dissipation is done by imposing variable loss rates in a single site of a one dimensional optical lattice hosting a Bose-Einstein condensate. The authors have the capability to readout in real time the losses of atoms thanks to an ion detection system that is the trademark of the group’s experiment, and is worldwide unique. The experiments are timely and highly relevant. Indeed, the interplay of dissipation, interactions and quantum coherence is a important emerging field, and the authors unique experimental capabilities have shown in the past to deliver important insights.

The experiment builds upon the recognized track record of the group studying the same system under different conditions. It is very close both in the scope and spirit to a recent paper of the group (New J. Phys. 24 103034, 2022). The novelty of the study presented here is twofold: on the one hand, the strength of the dissipation is used as a controlled parameter, rather than tunneling, and on the other hand data are presented regarding the effects of finite temperature on the hysteretic behavior. The use of dissipation strength is of interest because it is supposed to yield a different critical exponent for the phase transition, which to my knowledge has not been observed before. In this respect, figure 5a of the paper is the most important novelty. The temperature investigation is also new, although I am not sure how much insight can be drawn from it at that stage.

Even though the results are interesting and new and can potentially be worth a publication in SciPost physics, I do not think that the paper is suitable for publication in its current form. As I detail below, it suffers from presentation issues, and the interpretation is sometimes confusing and superficial. In my opinion, it needs in-depth rewriting:

1- The quality of writing is rather poor, with a number of very general, vague considerations that are not directly connected to the work presented, particularly in the introduction. A more concise and precise formulation should be preferred. An example is the digression on transport, ending with the rather cryptic sentence ‘Understanding dissipative phase transitions is therefore key to design and engineer transport processes in micro- and nanoscale quantum devices and to harness the system’s non-analytic behavior. ’ Another example is the second last paragraph of the introduction. In facts, the dissipation strength has been widely tuned in a number of experiments in the past, in the form of a controlled level of spontaneous emission (see for example Nature Physics 16, 21–25 (2020)), two-body losses in photoassociation (Science advances 3.12 (2017): e1701513.) to name a few. Even for cavities, for most relevant cases of dispersive coupling the effective dissipation induced by photon losses can be actually tuned by controlling detuning (see PRX Quantum 3, 020308 (2022) for a recent review), with mirror losses only setting an upper bound to the coherence of photon-assisted processes. The presentation of temperature as another ‘ways to dynamical [sic] change the dissipation or decoherence’ is misleading. Thermalization in a closed quantum system differs from coupling to the environment at the fundamental level, as numerous theoretical and experimental papers have shown in the last years (take for instance the huge literature on many-body localization, to name only one example).

2- I understand that the one dimensional lattice forces to consider the transverse degree of freedom in the analysis. However, the entire interpretation of the data relies on the single mode model, which seem to reproduce most of the qualitative features observed in the experiment. I do not see in the data any compelling evidence that the multimode nature of the system plays any significant role. The model not only disregards the multimode aspects but also a lot of other aspects beyond mean field, noise from the drive etc, that discrepancies with the experiment may have another origin.

3- I find the discussion of the atom number fluctuation also confusing: the excess noise indicates large fluctuations of the jumping time from one realization to the other, so that the correlations in the ion signal are classical. This was in fact very well described in the previous paper of the group studying the very same bistable system upon variations of the drive (New J. Phys. 24 103034, 2022), and the presentation there with a histogram of the jumping times is far more instructive.

4- I am also not completely sure I get the discussion concerning bunching and the value of g^2 being larger than 1 at some places: what does the ‘lower’ and ‘higher’ branch refer to ? When I look at the orange curve at low $\gamma$, I see the excess noise due to the random jumping sightly extending down to zero, I am not sure there is anything more in the signal.

5- In the calculation of the $g^2$ correlation function presented in figure 4b, I am wondering how the parameters for the single mode drive have been chosen, are they fitted on the data or taken from an estimate of the experimental parameters ? I am wondering also about the $g^2$ going below 1 in some parts of the blue curve, indicating strong non-classical correlations. Is that expected in some regime of parameters due to the strong Kerr effect ?

6- Figure 5 presents the most important result of the paper, the scaling of the hysteresis loop with the sweep rate is presented. Extracting these exponents is notoriously hard, and usually requires scanning the parameters over orders of magnitudes to obtain reliable data. I understand that it is hard to do in the experiment, but I would like at least to see on the graph the fit with \alpha=1 (the theoretical expectation) for comparison. I also think that the error bar of 5% certainly underestimated given the number of points (for sure if it is simply inferred from the non-linear fit error), so I would certainly not interpret the 10% disagreement with the simple theory as indicative of any fundamental deviation.

7- The temperature dependence presented in figure 6 is also new, but I am really not sure about the interpretation: upon decreasing the condensate fraction, I would expect the strength of the classical drive F to decrease, even at fixed J, because F directly represent the coherent state amplitude originating from the condensate in the neighboring lattice sites. Instead, there will be an incoherent drive due to thermal atoms tunneling, which has to be modeled by jump operators in the Lindblad equation. This is probably a simple extension of the formalism used so far in the paper. I would expect that trading coherent for incoherent drive would suppress the hysteresis without any need to invoke the multimode character of the system.

8- Temperature should be compared with other energy scales in the problem, like local mean field, level spacing, band width and gap etc in order to appreciate the origin of the disappearance of the hysteretic behavior. If temperature itself is irrelevant but condensate fraction is, then I would like the graph to be presented as a function of n_c (as the main axis). The sentence in the abstract pointing at a link between multimode system and finite temperature sounds more like a guess.

In addition, there are a few minor presentation issue that I would suggest the authors correct:

9- In most examples cited in the introduction (refs 8 an following) there is actually no dissipation, while the text suggests that they are examples of dissipative phase transitions. I would restrict to references that I directly relevant. Some of these references are rather arbitrary, such as [9], when the transition to optical instability in cold atoms cavity QED has been studied directly by the Esslinger and Stamper-Kurn groups, and by the Hemmerich in an earlier paper.

10- In figures 5 and 6 there are strange ‘[1]’ in the labels of axis, that should be removed

11- The sentence ‘Despite the functional difference, how the dissipation and the drive enter the equation of motion (see e.g. Eq.5) the general behavior is similar. ‘ needs reformulation.

12- In the section ‘Physical setup and theoretical description’, there is only one subsection appearing at the end of the section which is not standard.

13- In the conclusion ‘density-density fluctuations’-> density fluctuations

  • validity: -
  • significance: -
  • originality: -
  • clarity: -
  • formatting: -
  • grammar: -

Author:  Marvin Röhrle  on 2024-03-28  [id 4381]

(in reply to Report 1 on 2024-02-15)
Category:
answer to question

We would like to thank you for this extensive and insightful report on our paper.

Here are our answers to the concerns you have raised.

1 - We rewrote the third paragraph and onward in the introduction, in order to focus more on the actual results presented in the paper. This ties in to some other issues you have raised. In order to improve the writing we additionally clarified and rewrote large portions of the sections which you criticized in the other points (atom number fluctuations and temperature dependence, see the specific points for more information).

2 - We changed the focus of the paper to highlight more the qualitative similar features with the single mode model instead of trying to explain deviations with the multi-mode nature of the system. When introducing the single mode model, we added another citation to a more recent paper using a single mode model with qualitative agreement to previous results (Ceulemans et al., Phys. Rev. A 108, 013314 (2023)).

3 - In this case the jumping time also corresponds to a different dissipation strength, since it is dynamically ramped, i.e., the point at which a “critical” dissipation strength for switching is reached differs from one realization to another. The correlations calculated from the ion signal are indeed classical, however, the reason for this variation in times is due to the competition between the coherent drive and incoherent losses. We made the relation between the $g^2$ and dissipation strength more clear in the text.

4 - The "lower" branch is the part of the curve for higher dissipation strengths higher than the peaks, where the number of atoms in the site is lower. The "upper" branch is the of the curve for dissipation strengths smaller than the peaks, with large atom numbers. We added clarification to the text, which part of the curve this corresponds to and used different wording for the two parts. Furthermore, we added two guides to the eye in the figure to highlight the differences.

5 - The parameters are chosen to have a similar ratio as in the experiment, while showing qualitatively similar behavior. Yes, there is a parameter regime for the Kerr model to exhibit anti-bunching by converting phase fluctuations to intensity fluctuations (Rev. Mod. Phys. 54, 1061 (1982), Section V.C). We added a sentence noting this to the section.

6 - We added a fit for a fixed $\alpha=-1$ exponent in figure 5 (a). Furthermore, we replaced the explanation of the discrepancy with the multi-mode nature with: “However, the red dashed line in Fig. 5(a) with an exponent of $\alpha = −1$ is also an acceptable fit to the data. In order to get a better result, the hysteresis has to be measured for longer sweep times, which is not possible in our system, due to the constant atom loss.”

7 - Coupling the system to a thermal bath in the Lindbladian instead of coherently coupling to a condensate indeed cancels the hysteresis. We replaced the old argument with “ In fact, when replacing the coherent drive in Eq. 3 with a coupling to a thermal bath in Eq. 4 the hysteresis also vanishes in the single mode model. We conclude that only with coherent coupling to the neighboring sites the system exhibits hysteresis when varying the dissipation strength γ, while, when reducing the coherent fraction, the hysteresis area starts to disappear.” to reflect this observation.

8 - The temperature is T = 130 nK = 2700 Hz; the gap energy is around 18 kHz for the used lattice depth; $g n_\mathrm{peak}$ around 1 kHz; level spacing of harmonic oscillator in radial direction ~200 Hz, The thermal energy is larger than the local mean field energy and harmonic oscillator spacing, however, it is smaller than the gap energy. We have exchanged the x-axis for figure 6 as well as focusing the discussion on the condensate fraction, since the hysteresis requires a coherent drive in order to be seen as discussed in the answer to question 7. Furthermore, we rewrote the statement in the abstract to point towards this interpretation.

9 - These cited examples show hysteretic behavior, which is not necessarily around a first order dissipative phase transition. We split up this part and there is now one paragraph dealing with dissipative phase transitions, with new citations appropriate for the topic, and the other one with hysteresis in closely related systems, which contains the old citations 8 to 15.

10 - We removed the '[1]' unit labels, since the quantities are dimensionless.

11 - We removed the sentence, since it does not add value to the discussion at this point.

12 - We removed the subsection heading.

13 - We fixed this error.

---

## Round 1 · Referee Report · Anonymous (Referee 2) · 2024-3-23

Strengths

1)experimental investigation of first order dissipative phase transition with tunable dissipation strength
2)theory and numerical simulation complements experimental findings

Weaknesses

1)Description of model in Eq. (1-2) and the reduction to a single mode model in Eq.(3) could be better justified/connected to the experimental platform in Fig.1 . In particular it is not clear why and to what extent the rest of the system could be treated as just a classical drive for the lossy site, nor why the reduction to a single mode is justified. 2) Not clear what is experimentally measured to obtain the hysteresis loop in the Figures. A more detailed and explicit discussion of the experimental protocol could help the reader

Report

This work reports the experimental investigation of a dissipative phase transition in a system of ultracold bosonic atoms in a one dimensional optical lattice. The central site of the lattice is exposed to single particle losses, making the entire system open and dissipative. One key feature of this work is the possibility to experimentally tune the strength of this dissipation. Using this new knob the Authors study bistability and dynamical hysteresis under sweeping of the local dissipation strength, by measuring the atom number at the lossy site. The main result of this work, summarised in Fig. 5a, is the scaling of the hysteresis area with the sweep time, displaying a power law behavior. I believe this is a nice and important work that deserves publication. I have some comments/requested changes that I would like the Authors to take into account before publication (see above and below).

Requested changes

Expand and clarify the models in Eq.(1-2-3) and how they relate to the experimental platform (see comments above). Discuss more precisely what it is meausured and how it relates to the theoretical model.

  • validity: top
  • significance: high
  • originality: good
  • clarity: good
  • formatting: perfect
  • grammar: perfect

Author:  Marvin Röhrle  on 2024-03-28  [id 4382]

(in reply to Report 2 on 2024-03-23)

We would like to thank you for this helpful report on our paper.

Here are our answers to the concerns you have raised.

1) To address the first issue about a classical drive, we discuss in the section about temperature dependence the case of a thermal (incoherent) drive in the master equation and in this case the hysteresis and $g^2$ correlations vanish. Therefore, it is important that we have a coherent drive in the system. We added this to the section about the temperature as: “ In fact, when replacing the coherent drive in Eq. 3 with a coupling to a thermal bath in Eq. 4 the hysteresis also vanishes in the single mode model. We conclude that only with coherent coupling to the neighboring sites the system exhibits hysteresis when varying the dissipation strength γ, while, when reducing the coherent fraction, the hysteresis area starts to disappear.” When the theory model is introduced, we added a citation to another recent theoretical work (Ceulemans et al., Phys. Rev. A 108, 013314 (2023)), which also reduces the system to a single mode variant and it is in qualitative agreement with previous results. Based on this and the fact, that the single mode master equation model is the most faithful representation, while still being computational possible, we use it to make qualitative comparison with the experimental data.

2) We added a section to the supplementary material with an in-depth description of the experimental sequence and make an explicit reference in the main text for readers who would like to know more about it. It contains a new figure with a sketch of the sequence with details about the protocol. We hope, that this is helpful in understanding the experimental protocol of the measurements.

---

## Round 2 · Referee Report · Anonymous (Referee 2) · 2024-5-18

Report

Dear Editor,
I have read the Authors reply to previous reports and the new version of the manuscript. The main issue concerning the clarity of the presentation has been satisfactorily addressed in my opinion, the manuscript reads better in particular the Introduction, the presentation of the model and the discussion of the results which, as I had mentioned in my first report, are of high scientific quality. I think the paper deserves publication in SciPost Physics.

Recommendation

Publish (easily meets expectations and criteria for this Journal; among top 50%)

---

## Round 2 · Referee Report · Anonymous (Referee 1) · 2024-5-21

Report

With the improved presentation and discussion, the paper can be published.

Recommendation

Publish (meets expectations and criteria for this Journal)

---

## Round 2 · Author Response

Based on the feedback from the referees we rewrote some of the sections in order to make the presentation of the results more clear. Furthermore, we added some additional physical arguments, which address questions raised by the referees. We expanded the supplementary to explain the experimental sequence.

---

## Round 2 · List of Changes

• Rewrote the sentence in the abstract about the temperature dependence
  • Rewrote the introduction starting from the third paragraph, it first introduces the concept of dissipative phase transition and then emphasizes the progress of hysteresis in related systems and adding more citations to appropriate papers
  • We removed the single subsection heading in the "Physical setup and theoretical description" section
  • Added a reference in the experimental setup part to the supplementary material, which has a section with a detailed explanation of the experimental sequence and a graph visualizing it
  • Added another reference to a recent theoretical work, which uses a single mode model, in order to motivate our use of a single mode model for qualitative comparison
  • In the discussion of the correlations we made it more clear, what part of the graph we refer to and added guides to the eye to figure 4 (a)
  • We added a note about the $g^2$ of the theory curve going below 1
  • In figure 5 (a) we added a theory fit with $\alpha = -1$ and added discussion to the text, that this is also acceptable
  • We exchanged the x-axis in figure 6, the condensate fraction is now the main axis
  • We focus in the temperature section more on the condensate fraction and coherence, since the hysteresis disappears for a single mode model with thermal (incoherent) drive
  • We fixed errors in the labels of figure 5 and 6
  • We fixed a grammatical error in the conclusion
  • We changed the conclusion to reflect the changes in the other sections and introduction

---

## Editorial Decision

published